# Non-structural carbohydrates mediate seasonal water stress across Amazon forests

Caroline Signori-Müller [1,2✉], Rafael S. Oliveira [3], Fernanda de Vasconcellos Barros[4,5], Julia Valentim Tavares [2], Martin Gilpin[2], Francisco Carvalho Diniz[2], Manuel J. Marca Zevallos[6,7], Carlos A. Salas Yupayccana[7], Martin Acosta[8], Jean Bacca[6], Rudi S. Cruz Chino [7], Gina M. Aramayo Cuellar[9], Edwin R. M. Cumapa[6], Franklin Martinez[9], Flor M. Pérez Mullisaca [6], Alex Nina [7], Jesus M. Bañon Sanchez[6], Leticia Fernandes da Silva [8], Ligia Tello[6], José Sanchez Tintaya[6], Maira T. Martinez Ugarteche[9], Timothy R. Baker[2], Paulo R. L. Bittencourt[4,5], Laura S. Borma [10], Mauro Brum[11,5], Wendeson Castro [8], Eurídice N. Honorio Coronado [12], Eric G. Cosio[13], Ted R. Feldpausch [4], Letícia d'Agosto Miguel Fonseca[10], Emanuel Gloor[2], Gerardo Flores Llampazo[14], Yadvinder Malhi [15], Abel Monteagudo Mendoza[6], Victor Chama Moscoso[6], Alejandro Araujo-Murakami[9], Oliver L. Phillips [2], Norma Salinas[15,13], Marcos Silveira [8], Joey Talbot[16], Rodolfo Vasquez[17], Maurizio Mencuccini[18,19] & David Galbraith [2]

Non-structural carbohydrates (NSC) are major substrates for plant metabolism and have been implicated in mediating drought-induced tree mortality. Despite their significance, NSC dynamics in tropical forests remain little studied. We present leaf and branch NSC data for 82 Amazon canopy tree species in six sites spanning a broad precipitation gradient. During the wet season, total NSC ($NSC_T$) concentrations in both organs were remarkably similar across communities. However, $NSC_T$ and its soluble sugar (SS) and starch components varied much more across sites during the dry season. Notably, the proportion of leaf $NSC_T$ in the form of SS ($SS:NSC_T$) increased greatly in the dry season in almost all species in the driest sites, implying an important role of SS in mediating water stress in these sites. This adjustment of leaf NSC balance was not observed in tree species less-adapted to water deficit, even under exceptionally dry conditions. Thus, leaf carbon metabolism may help to explain floristic sorting across water availability gradients in Amazonia and enable better prediction of forest responses to future climate change.

A full list of author affiliations appears at the end of the paper.

Plants rely on both newly assimilated carbon and stored reserves of non-structural carbohydrates (NSC) for growth and other physiological functions such as respiration, osmotic regulation and defence[1,2]. As NSC stores reflect the balance of carbon supply via photosynthesis and its utilisation for plant metabolism, they are highly dynamic in time[3]. NSC stored during periods when supply exceeds demand are thought to constitute an important buffer during periods of environmental stress when carbon demand outstrips supply[2,4]. As a result of this, considerable attention has been paid to the potential role of stored NSC in mediating tree tolerance and survival under drought[5,6], during which stomatal conductance and assimilation rates are reduced to prevent water loss[7]. Experimental studies on both temperate[8] and tropical[9] seedlings have indicated an important role of NSC in the physiological mechanism of mortality, as plants with higher NSC content had higher survivorship under drought. However, the extent to which NSC metabolism moderates tolerance to water deficit in adult trees over large geographical domains remains unclear[10].

An understanding of the functional role of NSC in response to water deficit is of particular importance for the Amazon rainforest, the Earth's largest tropical forest region and a major terrestrial carbon sink, responsible for absorbing 5–10% of global anthropogenic $CO_2$ emissions[11]. Over the last 15 years, the Amazon has been subject to three large-scale drought events[12–14] and climate models project an intensification of drought risk over large parts of the Basin in the future[15]. Recent modelling results suggest that NSC play an important role in regulating the impacts of drought on carbon fluxes in the Amazon by maintaining growth under water deficit[16]. Moreover, observations of sustained stand-scale net primary productivity during the 2010 drought have also led to suggestions that Amazon trees deplete their NSC reserves during periods of water stress to prioritise growth[4]. However, empirical studies of NSC dynamics in tropical forests are rare, being limited to a small number of sites and species[17–19]. While a study in Panama found that NSC concentrations increased in the dry season[18], the only detailed community-level study in lowland Amazon forests, in a throughfall exclusion experiment in Eastern Amazonia, found that NSC reserves in trees subjected to long-term drought did not differ from those in unstressed trees[20]. Yet Amazon forests vary greatly in climate[21], soils[22] and plant life history strategies[23–25], of which could potentially influence NSC dynamics and forest response to climate change[1]. The scarcity of empirical data in Amazonia impedes understanding of the significance of NSC in modulating forest responses to water stress and thus limits current vegetation model development efforts to simulate drought impacts on tropical forests[16,26].

To address this significant data gap, we conduct a large-scale sampling of NSC across Amazon forests, using fully standardised field and laboratory protocols, performing all NSC analyses in the same lab (see Methods). Such standardisation is critical as differences in sampling and laboratory extraction protocols can yield substantial variation in NSC estimates, obstructing meta-analysis and comparisons across studies[27,28]. We analyse the concentration of NSC in leaves and branches of 82 canopy tree species in six sites across the Amazon Basin (Fig. 1; Table 1; Supplementary Table 1) effectively spanning the entire Amazon gradient in mean annual precipitation (1167–3155 mm year$^{-1}$; Fig. 1) and seasonality (0–7 months with rainfall $\leq 100$ mm month$^{-1}$) and including one site (Man) that experienced an atypically strong drought event[14]. We collect the plant material for NSC analyses in all sites during wet months (precipitation >100 mm month$^{-1}$), hereafter referred to as the wet season. For the four sites with more marked seasonality (Ken, Fec, Man and Tam), we also collect plant material in the peak of the dry season, where monthly precipitation is

$\leq 100$ mm month$^{-1}$ (Supplementary Fig. 1). We focus not only on total NSC concentrations ($NSC_T$) but also on the partitioning of NSC into its two major components: soluble sugars (i.e., oligosaccharides such as glucose, sucrose, fructose, etc.) and starch, as these fulfil distinct roles in plants[26]. Soluble sugars (SS) provide an immediate energy substrate for respiration, defence, plant stress signalling, phloem transport and osmoregulation[1]. Starch represents a transient or long-term energy store that plants can convert to SS for use when C demand exceeds supply[29]. To characterise plant water status at the time of NSC sampling, we measure midday leaf water potential ($\Psi_{MD}$) in all sites during the dry season, and in the two sites without a climatological dry season (Alp and Suc). Community-level (mean value of all species in each site) $\Psi_{MD}$ ranges from $-0.62 \pm 0.05$ MPa (mean $\pm$ SE) in the everwet Suc and Alp sites to $-2.18 \pm 0.30$ MPa in the ecotonal Ken site with the longest dry season.

We use this multi-site dataset to gain insights into how NSC and water deficit responses are related across Amazonian forests. We evaluate how leaf and branch NSC vary with water availability, both in space and seasonally, and also the relative roles of taxonomy and environment in determining NSC concentrations. Based on ecosystem modelling results[16] and observations suggesting prioritisation of aboveground growth under drought in Amazonia[4], we hypothesise that the drier sites would experience more seasonality in NSC stores and would also have greater NSC stores in the wet season. Our analyses reveal an important role of non-structural carbohydrates, and soluble sugars in particular, in mediating responses to seasonal water stress in Amazonian forests (Abstract available in Portuguese and Spanish, Supplementary Notes 1, 2).

## Results and discussion

**Wet season (baseline) NSC: role of environment vs. taxonomy.** Despite the wide range of species sampled and the differences in species composition across our study sites, we found little variation in leaf and branch $NSC_T$ and its components across sites in the wet season (monthly precipitation > 100 mm month$^{-1}$). In fact, wet season $NSC_T$ and SS in both leaves and branches did not differ significantly across sites (Fig. 2a, e; Supplementary Fig. 2; Supplementary Table 2). Overall, our results support the vegetation modelling assumption of spatially invariant baseline community-level $NSC_T$[16], thus potentially simplifying modelling of NSC dynamics in Amazonia. Only leaf starch exhibited significant differences across sites in the wet period, being markedly lower in the moderately seasonal sites (Man and Tam; $p = 0.001$, Supplementary Fig. 2) than in the driest Ken site and the two wettest sites (Suc and Alp). The higher wet season leaf starch concentrations in these sites may represent important strategies for maintaining function under an extended period of depleted water availability[29] in the case of Ken or light limitation[30] in the case of Suc and Alp.

We find that nested family–genus–species identity is very important, explaining much more of the overall variation in wet season $NSC_T$ and SS than sampling site for both leaves and branches (Fig. 2b, f). Taxonomy was a particularly important control for leaf NSC, explaining 67% and 72% of the wet season variation in leaf $NSC_T$ and SS (Fig. 2b). Within individual sites, there were clear species-level differences in $NSC_T$ and SS, which varied by a factor of between 4 and 10 across species (Supplementary Table 3, Supplementary Fig. 3 & 4). Moreover, when species occurred across more than one site, they largely maintained similar wet season levels of NSC and its fractions across sites (Supplementary Figs. 5–8). Despite the large amount of variance explained by taxonomy, $NSC_T$ and its constituent fractions are generally not related to plant traits that are indicative of life history strategies[25], such as potential tree size, mean growth, mortality rates and wood density[19] (Supplementary

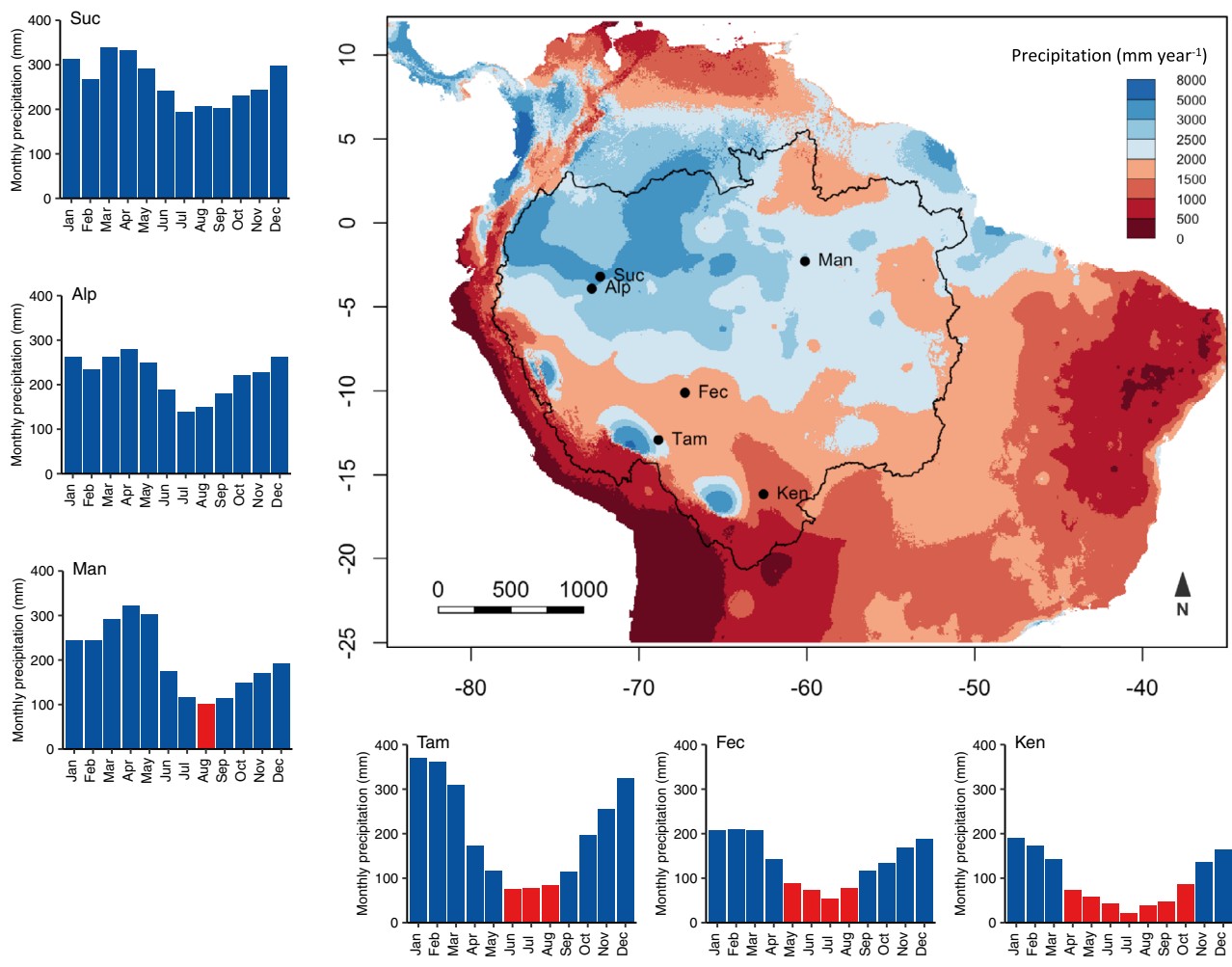

**Fig. 1 Location of sampled sites and monthly precipitation.** The map depicts mean annual precipitation in South America (mm year$^{-1}$). The Amazon basin is bounded by the black outline. Bar plots show the monthly precipitation for each site, blue represents precipitation >100 mm month$^{-1}$ and red ≤100 mm month$^{-1}$. Data for the map and bar plots are from WorldClim v2 (1970–2010, 30 s resolution)[51].

| Table 1 Site information. | | | | | | | |
|---|---|---|---|---|---|---|---|
| Site name | Short code | Site location | MAP (mm)[a] | DSL[b] (months) | $\Psi_{MD}$ (MPa)[c] | n | Collection period |
| Kenia | Ken | Ascensión de Guarayos, Santa Cruz, Bolivia (16°1'S, 62°43'W) | 1167 | 7 | −2.18 ± 0.81 | ~47% b.a[d] 9 species 25 trees | Dry: Jun 2017 Wet: Mar 2017 |
| Catuaba Experimental Farm | Fec | Senador Guiomard, Acre, Brazil (10°4'S, 67°37'W) | 1658 | 4 | −2.05 ± 0.51 | ~40% b.a 14 species 40 trees | Dry: Jul 2017 Wet: Apr 2017 |
| Cuieras Biological Reserve | Man | Manaus, Amazonas, Brazil (2°36'S, 60°12'W) | 2420 | 1 | −1.58 ± 0.40 | ~12% b.a 13 species 34 trees | Dry: Oct 2015 Wet: Jun 2016 |
| Tambopata National Reserve | Tam | Puerto Maldonado, Madre de Dios, Peru (12°49'S, 69°16'W) | 2451 | 3 | −1.10 ± 0.36 | ~40% b.a 21 species 58 trees | Dry: Jul 2017 Wet: Feb 2017 |
| Allpahuayo Mishana National Reserve | Alp | Iquitos, Maynas, Peru (3°56'S, 73°25'W) | 2660 | 0 | −0.64 ± 0.26 | ~26% b.a 27 species 85 trees | Oct 2017 |
| Sucusari Ecological Reserve | Suc | Iquitos, Maynas, Peru (3°15'S, 72°54'W) | 3155 | 0 | −0.62 ± 0.33 | ~21% b.a 31 species 93 trees | Nov 2017 |

[a]MAP mean annual precipitation (mm) from WorldClim Bioclimatic variables version 2, 30-s resolution[51].
[b]DSL dry season length, number of months with precipitation ≤100 mm, data extracted from WorldClim Version 2, 30 s resolution[51].
[c]$\Psi$MD = midday community leaf water potential in the sampled day during dry season ± SD (except in the sites where there is no meteorological dry season, ALP and SUC).
[d]b.a. = basal area, referring to the sampled percentage.

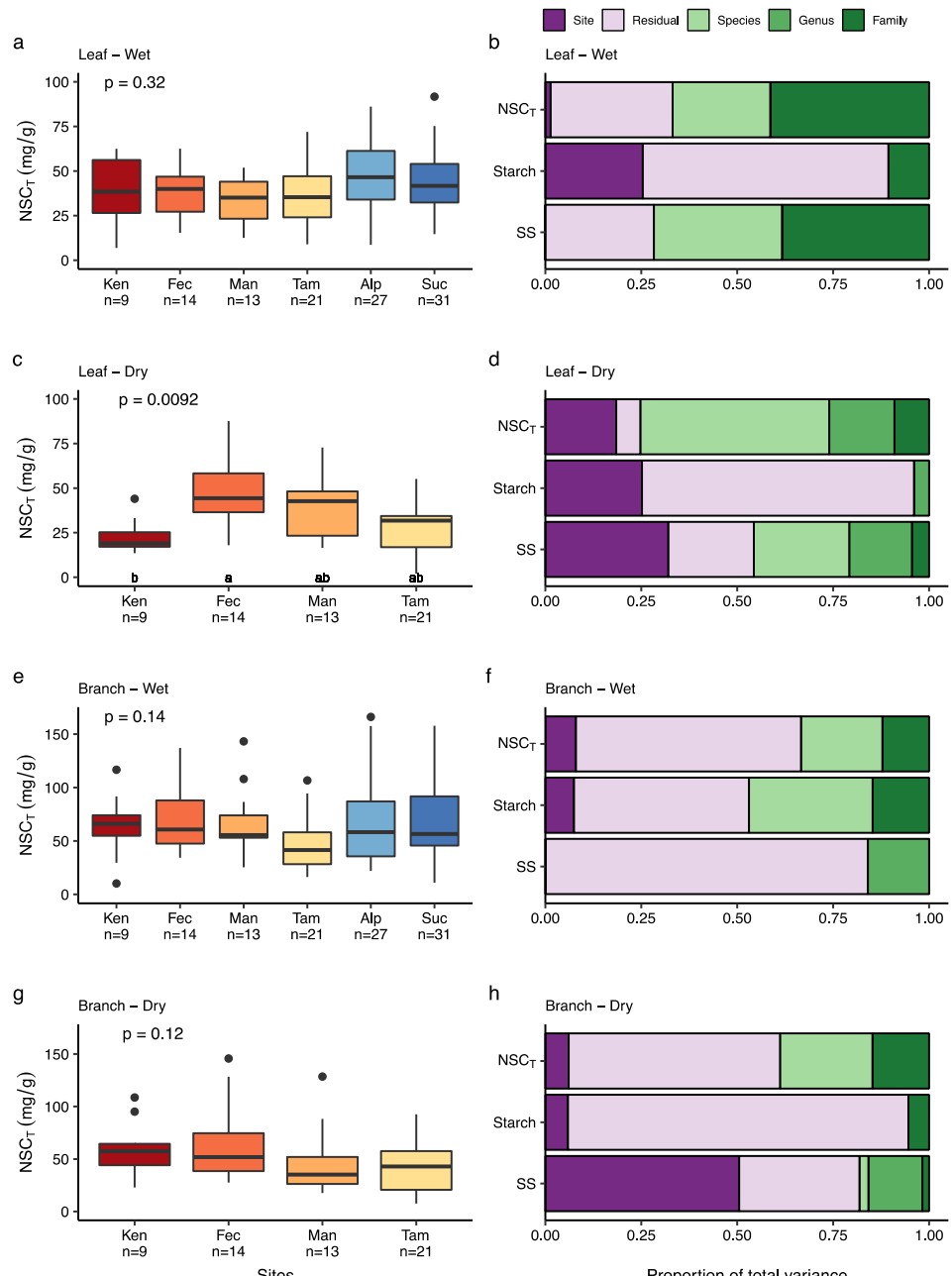

**Fig. 2 Species-mean total NSC (NSC$_T$) and variance partitioning into site and taxonomic components.** Concentrations of NSC$_T$ and variance partitioning results are displayed separately for leaves (panels **a**–**d**) and branches (panels **e**–**h**) during the wet and dry seasons. Left hand panels: Each box encompasses the 25th to 75th percentiles; the median is indicated by the horizontal line with each box while external horizontal lines indicate the 10th and 90th percentiles; dots indicate outliers. Sites are ordered and colour-coded from left to right from driest to wettest; red to yellow boxes represent the seasonal sites and two blue boxes the aseasonal sites; $n$ indicates the number of species sampled in each site. Differences among sites were tested using Kruskal–Wallis. Sites with different letters are statistically distinguishable ($p < 0.05$, post hoc Mann–Whitney–Wilcoxon Rank Sum test using Bonferroni correction is indicated by small letters). Right hand panels: Partitioning of total variance of NSC$_T$, starch and soluble sugars into genetic (family/genus/ species), environmental (site) and error (residual) components; for the variance partitioning analysis, values were log1p-transformed prior to analysis.

Figs. 9 and 10). Although our data on plant phenological strategies were limited, we do find that evergreen species have higher total leaf NSC$_T$ ($p = 0.023$) and SS ($p = 0.012$) in the wet season than semideciduous/deciduous species (Supplementary Figs. 11 and 12; Supplementary Table 4). The strong taxonomic influence on leaf NSC$_T$ and SS likely further relates to physiological attributes for which we have little current information. Leaves have intense metabolic requirements due to their roles in photosynthesis and phloem loading[31] and face greater

osmoregulatory and defence demands than other plant organs[32]. Thus, differences across taxa in photosynthetic and respiration rates as well as osmoregulatory and defence mechanisms, although little studied, may help to explain the strong taxonomic signatures we find.

**NSC seasonality.** NSC$_T$ exhibited greater differences across sites in the dry season compared to the wet season (Fig. 2c, g). This

was especially the case for leaf NSC, where $NSC_T$, SS and starch exhibited significant differences across sites, while in branches, only SS varied across sites (Fig. 2 and Supplementary Fig. 2). In line with this, our variance partitioning analysis also showed that site accounted for more of the overall variation in NSC metrics in the dry season than it did in the wet season (Fig. 2d, h).

Seasonal patterns of NSC across sites diverged markedly (Fig. 3; Supplementary Table 5; Supplementary Figs. 13 and 14). In the two driest sites in our network, Ken and Fec, we find strong evidence of mobilisation of starch reserves to SS. In Ken, the driest site evaluated, leaf starch reserves declined by 81% ($p < 0.001$) in the dry season while leaf SS concentrations remained unchanged ($p = 1$), despite a 43% reduction in leaf $NSC_T$ (Fig. 3, $p = 0.019$). In Fec, the second driest site evaluated, leaf starch concentrations also decreased markedly in the dry season (72% reduction; $p < 0.001$). However, in this site, significant increases in leaf (32% increase; $p < 0.001$) and branch (48% increase; $p < 0.001$; Fig. 2) SS were observed, with an overall increase in leaf $NSC_T$ in the dry season. The reduction of leaf $NSC_T$ in Ken but not in Fec may be attributed to stronger source limitation in the driest Ken

site (Supplementary Fig. 15)[33]. In the two less water-limited sites for which we had data in both seasons (Man and Tam), we interpret the seasonal dynamics of $NSC_T$ and its fractions to be driven mainly by growth. In the Man site, productivity is maintained at high levels during the dry season[34] (Supplementary Fig. 5), and the observed depletion of branch starch (60% reduction; $p < 0.001$) accompanied by an increase in SS concentrations (30% increase; $p = 0.001$) may be associated with enhanced branch growth[35]. In Tam, however, the dominant pattern was one of greater leaf $NSC_T$ and SS in the wet season (Fig. 3), which we attribute to higher productivity in the wet season, as observed by in situ NPP measurements[36] and also seen in MODIS-derived Enhanced Vegetation Index (EVI) values (Supplementary Figs. 15 and 16).

**Relationship between leaf NSC and water potential.** Despite the widely varying seasonal patterns across sites, we find a strong relationship between $\Psi_{MD}$ measured in the driest period of the year and the proportion of leaf $NSC_T$ allocated to SS (SS:$NSC_T$)

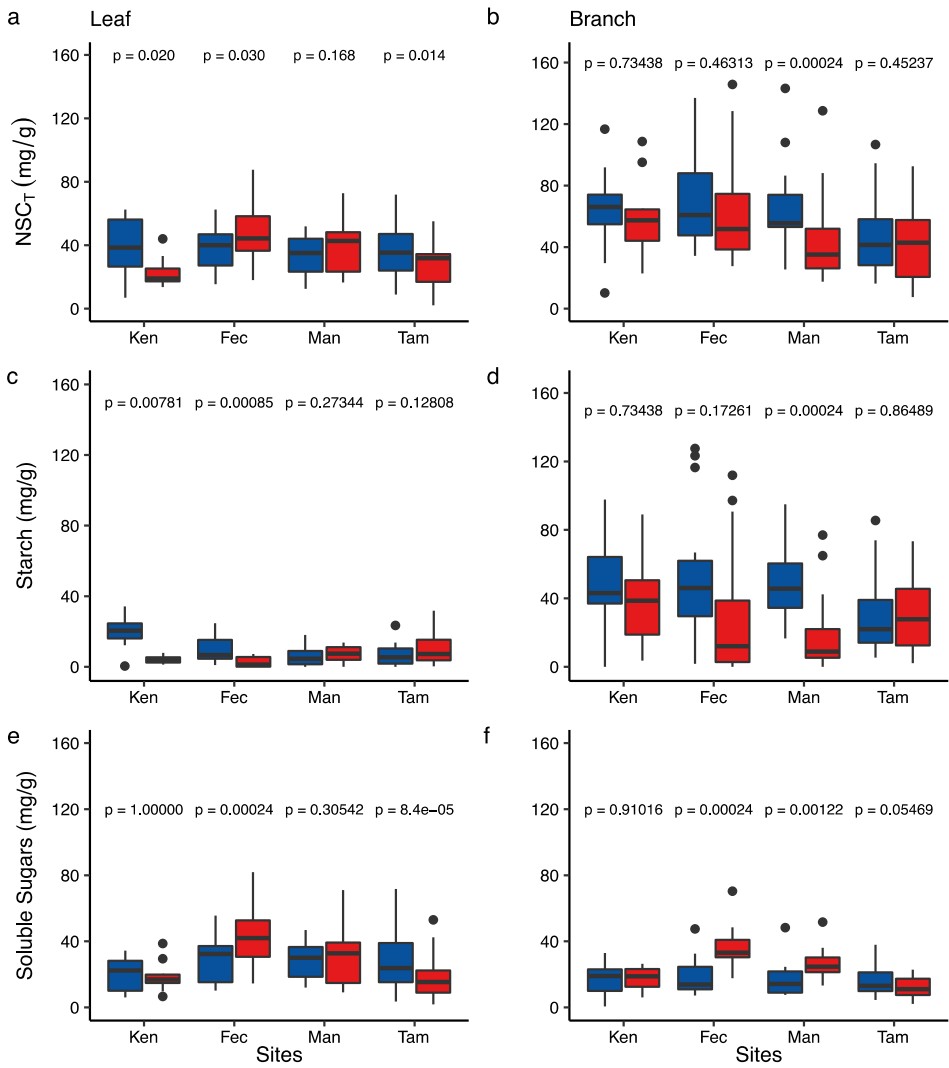

**Fig. 3 Seasonal variation of total NSC ($NSC_T$), starch and soluble sugars (SS) across Amazon forests.** Sites are ordered from left to right, from driest to wettest site. Panels **a**, **c** and **e** represent leaves and panels **b**, **d** and **f** represent branches. Red boxes denote the dry season and blue boxes denote the wet season. Each box encompasses the 25th to 75th percentiles; the median is indicated by the horizontal line with each box while external horizontal lines indicate the 10th and 90th percentiles; dots indicate outliers. Number of species sampled in each site are the same in the dry and wet season and are as follows: Ken = 9, Fec = 14, Man = 13, Tam = 21. To test for differences between season within site we used paired sample Wilcoxon tests.

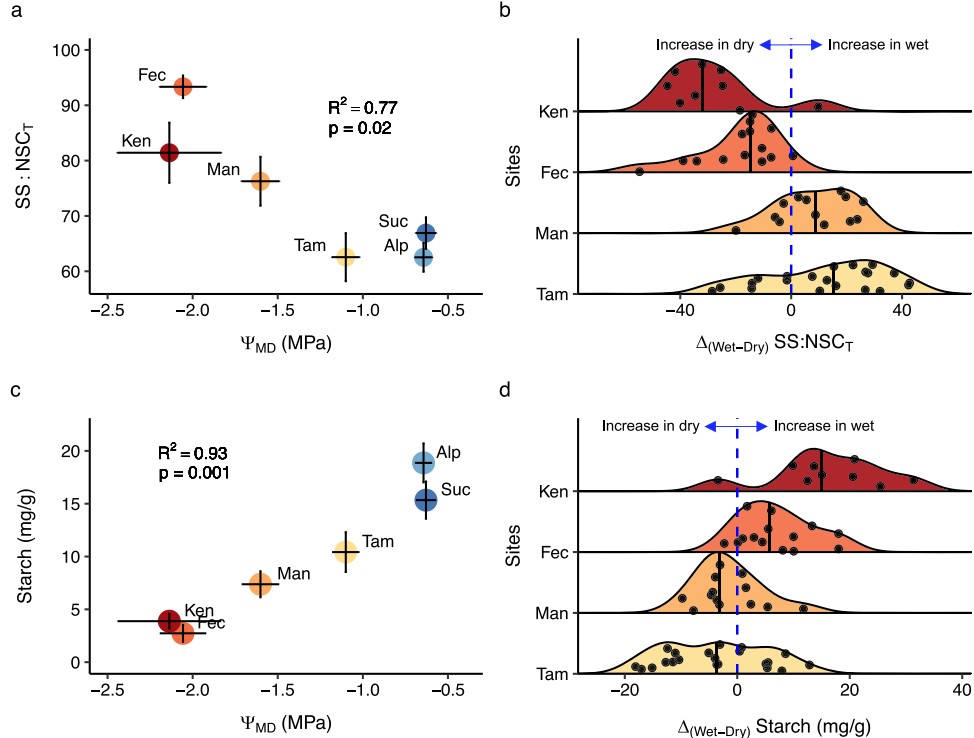

**Fig. 4 Relationship between leaf NSC and water status, and distributions of species-level seasonal shifts in leaf NSC allocation.** Relationship between **a** leaf SS:NSC$_T$ (proportion of leaf NSC$_T$ in the form of soluble sugars) and midday leaf water potential ($\Psi_{MD}$); **c** leaf starch and $\Psi_{MD}$. Distributions of species-level seasonal shifts in **b** leaf SS:NSC$_T$ allocation (SS:NSC$_{T\ Wet}$ − SS:NSC$_{T\ Dry}$) and **d** starch allocation (starch$_{wet}$ − starch$_{dry}$). In the panel **a** and **c** the SS:NSC$_T$, starch and $\Psi_{MD}$ represent mean of all species sampled in each site in the dry season, except in the two sites where there is no dry season (monthly precipitation ≤100 mm month$^{-1}$). Vertical and horizontal bars denote one standard error of the mean. The relationship between SS:NSC$_T$, starch and $\Psi_{MD}$ was fitted using standardised major axis (SMA) regression. In the panel **b** and **d** sites are ordered from top to bottom from drier to wettest. Long vertical black line denote the mean $\Delta$SS:NSC$_T$ and starch, each dot represents one species. Values to the left of the blue line denote species that increased SS:NSC$_T$ and starch in the dry season while those to the right of the line increased SS:NSC$_T$ and starch in the wet season.

(Fig. 4a; $p = 0.02$, $R^2 = 0.77$). This relationship also exists when taking predawn leaf water potential ($\Psi_{PD}$) as a measure of water status and at the species level as well as the community-level (Supplementary Fig. 17) and is underpinned by a strong decline across sites in leaf starch concentrations with increasing water deficit (Fig. 4c; $p = 0.001$, $R^2 = 0.93$). This relationship points to an notable increase of SS relative to starch in leaves during periods of water stress and is reinforced by the fact that in the two driest sites we found that almost all species increased SS:NSC$_T$ in leaves during the dry season (Ken: $p = 0.007$, Fec: $p < 0.001$; Fig. 4b; Supplementary Figs. 18 and 19). Indeed, in several species in Ken and Fec, dry season leaf starch reserves were effectively exhausted (zero or very close to zero) while leaf SS levels were unchanged (Ken) or increased (Fec). Furthermore, species co-occurring in both Fec and in the more mesic Tam site almost all had higher dry season foliar SS and lower foliar starch content in the drier site (Supplementary Figs. 20 and 21). However, we do not find evidence of enhanced conversion of foliar starch to SS in Man (Fig. 4b), a mesic site where species are less-adapted to prolonged water deficit, despite sampling during one of the most intense droughts on record at that site (Supplementary Fig. 1).

The increase of foliar SS relative to starch in the driest sites (Fig. 4b) and its strong relationship with community-level water status (Fig. 4a) suggest an important role of foliar SS in mediating responses to water deficit in Amazonian forests. The increased allocation of leaf NSC to SS in these sites is further independent of the seasonal behaviour of NSC$_T$ (Supplementary Fig. 22) and thus does not simply result from changes in source-sink dynamics. Studies mainly on agricultural crop systems[37–39] and

on a small number of shrubs[40] and trees[41] have shown that foliar SS can be very important for osmotic adjustment, actively accumulating in response to declining water potentials and thus helping plants to avoid dessication[42]. Our study suggests this phenomenon is widespread and that leaf SS contributes to the maintenance of hydraulic function in adult tropical trees across a broad range of taxa. A key future focus of research should be the identification of the specific sugars involved in osmoregulation in tropical plants. Studies on herbaceous species and temperate tree species suggest an important osmoregulatory role of oligosaccharides such as raffinose and pinitol[43–45], but their role in osmoregulation in tropical forest trees is unknown.

The ability of species to rapidly mobilise leaf starch into SS under water deficit is likely an important mechanism for tolerating water stress[46]. While our study shows that species found in drier forests of the Amazon almost all have this capacity, it is far from clear that species in less seasonal regions of the Amazon are able to adjust their SS balance to the same extent. Indeed, the lack of seasonal adjustment in leaf NSC allocation to SS under exceptional water stress in the mesic Man site suggests that tree species that are less adapted to strong seasonal drought may not have the capacity to rapidly adjust foliar SS under water stress. This capacity may ultimately be an important determinant of future Amazon forest composition under continued climate change.

## Methods
**Sites and species**. Plant material for NSC analysis was collected in six Amazonian sites (Fig. 1 and Table 1). These sites were selected from the RAINFOR network of

permanent and well-identified forest plots[12,47,48] as representing the Amazon-wide gradient in mean annual precipitation (MAP) and seasonality (Supplementary Fig. 22 and 23). The study sites also encompass a broad range of soil types and forest plant communities[22,49]. Our sites included aseasonal forests with no climatological dry season (no months with rainfall ≤100 mm[50]; Alp, Suc), forests with a moderate dry season (1–4 months with rainfall ≤100 mm; Fec, Man, Tam) and a transitional forest in the most seasonal site at the southern margin of the Amazon biome (7 months with rainfall <100 mm; Ken)[51]. Sampling took place during what were climatologically normal years in most sites, with the exception of Man, where sampling took place during the strong 2015 El Niño drought event (Supplementary Fig. 1). In total we sampled 82 canopy tree species, from 63 genera and 29 families (Supplementary Table 1). In each site, sampling was focused on the most dominant canopy species in terms of basal area, with the total number of species sampled at each site ranging from 9 to 31 (Table 1). Species-level identification of all trees sampled is based on botanical vouchers previously collected and deposited in Amazon state herbaria (AMAZ, CUZ, HOXA, INPA, UFACPZ, USZ) by RAIN-FOR partners. All plot trees are tagged and identifications are obtained from the ForestPlots.net database[48] (https://www.forestplots.net/; Ken, Fec, Tam, Alp and Suc) and from collaborator databases for Man[24]. All branches collected had fully expanded leaves, no evidence of liana infestation or injuries and were not shaded. Species collected in the wet season that did not keep their leaves during the dry season were excluded from all analysis to avoid potential biases due to different phenological strategies[1].

**Non-structural carbohydrate (NSC) sampling and analysis.** Two to six individuals were sampled per species, with all individuals being >20 cm in diameter at breast height (DBH). To minimise effects of diurnal changes in NSC concentrations, samples were obtained before sunrise in all sites, except Man. In Man, branches and leaves were obtained just after sunrise and always before 8 a.m. Leaf and branch samples were obtained by a tree climber from first order fully sunlit branches with fully expanded leaves and kept in ice during sampling and transported to the laboratory. Upon arrival at the laboratory, samples were microwaved for 90 s at 700 W to stop enzymatic activity that would otherwise affect NSC levels, and oven-dried at ~60 °C for at least 48 h or until they were completely dry (no >72 h). All NSC sample preparation and analyses were performed at the University of Campinas, in the laboratory of plant ecophysiology. Prior to NSC quantification, samples were ground to a fine powder (Geno/Grinder® SPEX SamplePrep mill). Branch samples had their bark removed before being ground.

Non-structural carbohydrates (NSC) are defined here as free, low molecular weight sugars (i.e., oligosaccharides such as glucose, fructose, sucrose, etc.) plus starch. NSC was analysed as described in Hoch et al.[52] with minor modifications. Two replicates of each sample were analysed and the mean of the two replicates used as the sample NSC value. First, we diluted ~15 mg of the ground plant material with 1.6 mL of distilled water and then incubated in a water bath at 90–100 °C for 60 min to solubilise sugars. Then we took an aliquot of 700 µL from each sample (700 µL). We use the remaining aliquot volumes (900 µL) to determine the SS concentration using invertase from *Saccharomyces cerevisiae* (Sigma-Aldrich, St. Louis, MO, USA) to break down sucrose and fructose to glucose[20]. Additionally, for both reaction routines, we used GAHK (Glucose Assay Hexokinase Kit - Sigma-Aldrich, St. Louis, MO, USA) together with phosphoglucose isomerase from *Saccharomyces cerevisiae* (Sigma-Aldrich, St. Louis, MO, USA). The concentration of free glucose was measured photometrically in a 96-well microplate spectrophotometer at 340 µm (EPOCH-Biotek Instruments INC-Winooski, VT-USA). The aliquot that we initially separated was incubated overnight to react with Amyloglucosidase from *Aspergillus niger* (Sigma-Aldrich, St. Louis, MO, USA) to breakdown the total NSC to glucose. Thereafter total glucose (corresponding to NSC) was determined as described above and starch was calculated as total NSC minus soluble sugars. All NSC values are expressed in mg/g dry mass.

**Predawn ($\Psi_{PD}$) and midday ($\Psi_{MD}$) leaf water potential.** In situ predawn and midday leaf water potential ($\Psi_{PD}$ and $\Psi_{MD}$ respectively) measurements were made in the same day on the same trees for which we obtained samples for determination of NSC concentrations. We sampled 2–6 trees per species and measured the $\Psi_{PD}$ and $\Psi_{MD}$ in 2–5 canopy fully expanded leaves using a pressure chamber (PMS 1505D and PMS 1000, PMS instruments) and the values were then averaged per individual. $\Psi_{PD}$ measurements were taken before sunrise from 3:30–5:30 and $\Psi_{MD}$ from 11:00 am –2:30 pm. Water potential data collection took place in what is typically the driest time of the year in each sampling plot except in the Alp and Suc sites where there is no climatological dry season (months with precipitation ≤100 mm) and little seasonality in rainfall. Owing to logistical limitations, we did not measure $\Psi_{PD}$ in the Man site.

**Enhanced vegetation index (EVI).** To gain further insights into how seasonality in canopy productivity might affect our observations, we extracted Enhanced Vegetation Index (EVI) values for each site derived from the MODIS-MAIAC product, using data from 2003 to 2018[53]. The surface reflectance data were normalised to nadir target and 45-degree solar zenith angle through the Bidirectional

Reflectance Distribution function, at a spatial resolution of 1 km and aggregated to biweekly (16-day) composites using the median values in this product, before EVI calculation[54,55]. The EVI was calculated using Eq. 1:

$$EVI = 2.5*\frac{\rho NIR - \rho Red}{\rho NIR + (6*\rho Red - 7.5*\rho Blue) + 1} \qquad (1)$$

where $\rho NIR$ is infrared reflectance, $\rho Red$ is red reflectance, and $\rho Blue$ is blue reflectance. The constants (6, 7.5, 1, and 2.5) in the divisor represent the aerosol coefficient adjustment of the atmosphere for the red and blue band, the adjustment factor for the soil and the gain factor, respectively[55,56].

The composites were retrieved considering only cloud-free and low atmospheric turbidity according to MAIAC quality flags. Further information on image processing and correction are described in Dalagnol et al.[54]. MODIS pixel values were extracted considering the coordinate system of each site using raster[57] and rgdal[58] R packages.

**Statistical analysis.** We performed all statistical analysis in R (R Core Team 2018, version 3.6.2)[59]. Preliminary tests included: analysis of normality (Shapiro–Wilk), homogeneity of variances (Fligner–Killeen) for each NSC fraction (NSC$_T$, SS and starch) in each organ (branches and leaves). As NSC$_T$, starch and SS were not normally distributed, these parameters were log1p-transformed[60] prior to variance partitioning. For comparison of means across sites and seasons, data were not log1p-transformed and non-parametric tests were used.

To evaluate differences across sites in NSC$_T$, SS and starch concentrations, a Kruskal–Wallis test was used (R base function). We conducted statistical analyses separately for each plant organ (leaves and branches) and season (wet and dry). When a significant site effect was found, a post hoc Mann–Whitney–Wilcoxon Rank Sum test using Bonferroni correction (from "Agricolae" package in R[61]) was performed to evaluate which sites were significantly different. To evaluate whether there were significant differences in NSC$_T$, starch and SS between seasons, we performed paired sample Wilcoxon tests (R base package) for each site and plant organ separately. The ALP and SUC sites were excluded from the seasonal analysis, as these sites were only sampled in one point in time. Figures were constructed using the "ggplot2" package[62] and to display the p-values in the figures we used the "ggpubr" package (stat_compare_means function)[63].

We conducted standardised major axis (SMA) regression, using the "smatr" package in R[64] to assess relationships between NSC$_T$, SS and starch and site level water-status ($\Psi_{min}$ and $\Psi_{PD}$) at both the community-level (mean species value per site) and species level. To account for differences across sites in seasonal source limitation we also tested for relationships between the ratio of SS and NSC$_T$ (SS: NSC$_T$) and $\Psi_{MD}$ and $\Psi_{PD}$. SMA regressions were conducted for each organ separately, using NSC values corresponding to the same season $\Psi$ was measured. We further tested for bivariate relationships between species-mean NSC metrics and plant attributes indicative of species life history strategies, including branch wood density (Tavares et al. in prep and Barros et al. unpublished data), mean growth rate, potential maximum size and mean mortality rate[25,65] using the lm function (R base package).

To determine the relative importance of taxonomy vs. measurement site in determining NSC$_T$ and its fractions, we undertook a variance partitioning analysis as described by Fyllas et al.[23], where a multilevel model was first fitted for each NSC fraction, organ and season according to Eq. 2:

$$T = \mu + p + f/g/s + \varepsilon \qquad (2)$$

where $\mu$ is the overall mean species value of each NSC fraction (T), p is the random site effect, i.e., the effect of the location at which each individual was found (soil and climate), f/g/s represents the random effect caused by the genetic structure of the data, i.e., that each individual belongs to a species (s), nested in a genus (g), nested in a family (f), and $\varepsilon$ is the residual term, which includes both the within-species variability not explained by site, as well as any measurement error. All parameters were estimated by the Residual or Restricted Maximum Likelihood (REML) method with the "lme4" package in R[66].

**Reporting summary.** Further information on research design is available in the Nature Research Reporting Summary linked to this article.

## Data availability

The non-structural carbohydrates concentration data are available at www.forestplots.net/data-packages/Signori-Muller-et-al-2021 (https://doi.org/10.5521/forestplots.net/2021_3)[67], all recorded species, genus and family names were checked and standardized using the Taxonomic Name Resolution Service (tnrs.biendata.org)[68]. The mean growth rate, potential tree size and mortality rate are available at www.forestplots.net/data-packages/coelho-de-souza-et-al-2016 (https://doi.org/10.5521/FORESTPLOTS.NET/2016_4)[66]. The climatic data are available at www.worldclim.org/data/index.html[51]. Enhanced vegetation index data are available at www.zenodo.org/record/3159488#.YBW_u3f7Tlw (https://doi.org/10.5281/ZENODO.3159488)[53,54]. Leaf water potential data are available upon reasonable request to the correspondence author. Branch wood density data are from Tavares et al. (in prep) and Vasconcelos Barros (unpublished data). The inventory data for species selection are from the RAINFOR network available upon request at www.forestplots.net[47,48].

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

## Acknowledgements

Amazon fieldwork and laboratory analyses for this publication were supported by the UK Natural Environment Research Council project TREMOR (NE/N004655/1) to D.G. This paper is an outcome of C.S.-M.'s doctoral thesis in the Plant Biology Programme at University of Campinas. C.S.-M., J.V.T., F.B.V., M.B., P.R.L.B., L.d'.A.F., L.S.B. and R.S.O. were financed by Coordenação de Aperfeiçoamento de Pessoal de Nível Superior-Brasil (CAPES Finance Code 001). C.S.-M. received a scholarship from the Brazilian National Council for Scientific and Technological Development (CNPq 140353/2017-8) and CAPES (science without borders 88881.135316/2016-01). D.G. acknowledges further support from the NERC-funded ARBOLES project (NE/S011811/1). J.V.T. acknowledges CAPES for its science without borders scholarship (99999.001293/2015-00). C.S.-M. and R.S.O. acknowledge the São Paulo Research Foundation (FAPESP-Microsoft 11/52072-0). M.M. acknowledges support from MINECO FUN2FUN (CGL2013-46808-R) and DRESS (CGL2017-89149-C2-1-R). R.S.O., F.V.B., L.S.B. and P.R.L.B. thank the U.S. Department of Energy, project GoAmazon (FAPESP 2013/50531-2). P.R.L.B. acknowledges The Royal Society for its Newton International Fellowship (NF170370). A number of the field sites (KEN, TAM, ALP) are part of the Global Ecosystems Monitoring (GEM) network (gem.tropicalforests.ox.ac.uk) and were supported by the Gordon and Betty Moore Foundation and by ERC Advanced Investigator Grant (GEM-TRAITS, 321131) to Y.M. The Amazon forest plots in the RAINFOR network analysed here were established, identified and measured with support from many colleagues and grants mentioned elsewhere[11,69]. This study was carried out as a collaborative effort of the ForestPlots.net meta-network, a cyber-infrastructure initiative developed at the University of Leeds that unites contributing scientists and their permanent plot records from the world's tropical forests. This paper is an outcome of ForestPlots.net approved Research Project #18. We additionally thank: Vanessa Hilares and the Asociación para la Investigación y Desarrollo Integral (AIDER) for field campaign support in Peru; Ezequiel Chavez and Noel Kempff Natural History Museum for field campaign support in Bolivia; and Hugo Ninantay and Alex Ninantay for sample collection.

## Author contributions

D.G., C.S.-M. and R.S.O. designed the study. C.S.-M. led the data analysis with inputs from D.G. and M.B. C.S.-M and D.G. wrote the manuscript with significant inputs from R.S.O. and M.M. J.V.T. and C.S.-M. led the field sampling in all sites, except MAN, which was led by F.V.B. C.S.-M., F.V.B. and M.G. performed the NSC analysis. C.S.-M., J.V.T., M.G., F.C.D., M.J.M.Z., C.A.S.Y., M.A., J.B., R.S.C.C., G.M.A.C., E.R.M.C., F.M., F.M.P.M., A.N, J.M.B.S., L.F.d.S., L.T., J.S.T. and M.T.M.U. collected the samples from all sites except for MAN where it was sampled by F.V.B., P.R.L.B., R.O. and L.S.B. T.R.B, W.C., E.N.H.C., E.G.C., T.R.F., E.G., G.F.L., Y.M., A.M.M., V.C.M., A.A.-M., O.L.P., N.S., M.S., J.T. and R.V. led the forestplot.net field expeditions for data collection. L.S.B. and L.d'A.M.F. provided the enhanced vegetation index data. All authors critically revised the manuscript and gave final approval for publication.

## Competing interests

The authors declare no competing interests.

## Additional information

[1]Department of Plant Biology, Institute of Biology, Programa de Pós Graduação em Biologia Vegetal, University of Campinas, Campinas, Brazil. [2]School of Geography, University of Leeds, Leeds, UK. [3]Department of Plant Biology, Institute of Biology, University of Campinas, Campinas, Brazil . [4]Geography, College of Life and Environmental Sciences, University of Exeter, Exeter, UK. [5]Department of Plant Biology, Institute of Biology, Programa de Pós Graduação em Ecologia, University of Campinas, Campinas, Brazil. [6]Universidad Nacional de San Antonio Abad del Cusco, Cusco, Peru. [7]Pontificia Universidad Católica del Perú, Lima, Perú. [8]Programa de Pós-Graduação em Ecologia e Manejo de Recursos Naturais, Universidade Federal do Acre, Rio Branco, Brazil. [9]Museo de Historia Natural Noel Kempff Mercado, Universidad Autonoma Gabriel Rene Moreno, Santa Cruz, Bolivia. [10]Earth System Science Centre, National Institute for Space Research, São José dos Campos, Brazil. [11]Department of Ecology and Evolutionary Biology, University of Arizona, Tucson, AZ, USA. [12]Instituto de Investigaciones de la Amazonia Peruana, Iquitos, Peru. [13]Sección Química, Pontificia Universidad Católica del Perú, Lima, Peru. [14]Universidad Nacional Jorge Basadre de Grohmann, Tacna, Peru. [15]Environmental Change Institute, School of Geography and the Environment, University of Oxford, Oxford, UK. [16]Institute for Transport Studies, University of Leeds, Leeds, UK. [17]Jardín Botánico de Missouri, Pasco, Peru. [18]CREAF, Universidad Autonoma de Barcelona, Barcelona, Spain. [19]ICREA, Barcelona, Spain. ✉email: carol.signori@gmail.com

