## [Peer Review File · Nature Communications]

REVIEWER COMMENTS

Reviewer #1 (Remarks to the Author):

This manuscript reports starch and simple sugar concentrations of terminal leaves and branches of 9-31 tree species from each of six Amazonian forests. Four of the sites are seasonal, with their dry seasons differing in severity and duration. In these 4 sites, leaves and terminal branches were sampled in both the wet and dry seasons. From two aseasonal sites without a dry season, sampling was done only during the wet season. This is a reasonable sampling design. The main scope was somewhat exploratory, but important because non-structural carbohydrates (NSC, the total of starch and simple sugar) are considered to be important for trees to cope with drought stress, which is widely recognized to be important for tree survival under the ongoing climate change. The manuscript summarizes a particularly nice dataset with broad geographical scope to allow generalization (six sites across Amazon), with standardized sampling.

The methods involving tree climbing to sample terminal and well-exposed branches from the top of the canopy, and careful laboratory measurements, appear rigorous. The statistical approach is solid. The results demonstrate some interesting and novel discoveries, including overall lack of differences among sites despite their differences in seasonality, as well as strong taxonomic signals in leaf simple sugar concentrations. The results shown in Figure 4 are particularly interesting to plant ecologists and physiologists, as they suggest community-level differences in adaptive syndrome of the use of simple sugars for osmoregulation. A hypothesis that plant survival under drought depends on NSC storage was popular about 10-20 years ago. However, more recent studies and perspectives do not support that survival is a simple function depending on NSC concentration in plants. The results shown here are consistent with this current view, and highlight not the energy storage per se, but what is detected as simple sugar concentration may be important for drought survival. The simple sugars involved in osmoregulation, however, are unlikely glucose, fructose, or sucrose, as they interfere with metabolism. Instead, other types of oligosaccharides are involved in preventing leaves from wilting. So, the work would be stronger if they could report the identity of the simple sugars that show differences among species, possibly with use of chromatography.

I find the very first sentence of the introduction to be a bit too ambiguous or too general. The question here is the energy storage, although it is impossible to separate what are NSC in active ongoing metabolism vs. storage. Starch and simple sugars have different functions with this regard.

Any species were shared among sites? In the variance partitioning, species are important, but if there was any species shared between two or more sites, how they differ among sites can be informative.

Leaf phenology can be important, but the manuscript does not mention anything about the phenology of the sampled species. Are they all evergreen species, and if so, when are the leaves produced and become fully expanded? Even in forests with limited seasonality, tree species show distinctive leaf production / replacement phenology.

Line 388 – simple sugars are expressed as glucose equivalent, but it may contain various sugars other than glucose, fructose, and sucrose. Especially, plant leaves use sugars other than these three for osmotic regulations.

To sum up, I like the manuscript very much, and overall the authors have done a great job in writing clearly. My main concern is that we don't know the identity of the simple sugars (SS) that show strong species and family-level signals (Figure 2), site differences (Figure 4a), and seasonal responses at the species-level (Figure 4b). I agree with the authors' interpretation that SS differences reflect species-level differences in the use of osmoregulating sugars (in leaf cell

vacuoles to avoid leaf wilting under water stress). But, the chemicals that cause species and seasonal differences in leaf SS concentrations are unlikely to be glucose, fructose, or sucrose. The authors may think that knowing their identity is beyond the scope of their work, but at least they have to discuss what are potential chemicals that act as osmoregulators.

Reviewer #2 (Remarks to the Author):

Comments on "Non-structural carbohydrates mediate seasonal water stress across Amazon forests"

General comments:

Non-structural carbohydrates (NSC) play an important role in tolerance to water stress and may mediate drought-induced mortality. This paper explored variabilities of total NSC and its components (soluble sugars and starch) across 83 canopy tree species in six Amazonian sites across a broad precipitation gradient. Total NSC concentrations in both leaves and branches were found similar across sites in wet season, but varied much more across sites during the dry season. At the community scale, leaf soluble sugars were found strongly linked to minimum leaf water potential. In drier sites, leaf soluble sugars increased during the dry season for almost all species, suggesting that leaf soluble sugars play a role of maintaining hydraulic status during water deficits in drier sites.

From the precipitation distribution map (Figure 1), as well as from Supplementary Figure 5 and 6, site Ken is the driest site with longest dry season, while Fec site range as the second driest site. However, the seasonal variation of total NSC, starch and soluble sugars (SS) showed opposite patterns, especially for leaves (Figure 3a, e, f), as dry season leaf NSC and SS were lower for Ken site, but high for Fec site as compared to those in wet season. At the less dry site Fec, absolute soluble sugars were higher than other sites for both wet and dry seasons, what are the possible reasons?

In the Results and Discussion (Line 215-220): The authors described these mixed results like: "In FEC, all species without exception increased the absolute leaf SS concentration in the dry season and all species except one depleted leaf starch stores. In KEN, dry season source limitation meant that several species did not increase absolute leaf SS concentrations and instead relied on larger depletions of leaf starch stores to maintain dry season SS at similar levels to the wet season (Figure 3)." Is this a site specific pattern? The opposite seasonal shift between Ken and Fec site reduce the general picture and make the results more difficulty to be explained.

Moreover, Figure 4 showed that Fec site always have the highest (see also Extended Data Figure 11) leaf and branch SS (% of leaf NSC corresponding to soluble sugars), but the distribution of of delta SS ($SS_{wet} - SS_{dry}$) at Fec site stay in between the driest site (Ken) and other wetter sites (Man and Tam). Is it possible use the delta SS (soluble sugar differences between wet and dry season) as a parameter for describing the relationships between leaf/branch water potential and NSC (total NSC, starch, and SS). Are there any relationships between leaf water potential and total NSC and starch concentrations?

Another concern is about the unbalance presentations of the data from full six study sites (named Ken, Fec, Man, Tam, Alp, Suc). From Figure 2, Figure 3 and Extended Data Figures 2, it seems that the experimental design did not consider sampling and NSC measurements for the wetter sites (Alp and Suc) during the dry season (Figure 2c, g; Figure 3, Extended Data Figure 2 c, d, g, h). In the Methods section, the authors did not indicated that NSC sample were not collected and measured for Alp and Suc sites during the dry season. Moreover, in the Methods section Lines 411-413, the authors indicated that "Water potential data collection took place in what is typically the driest time of the year in each sampling plot except in the ALP and SUC sites where there is no climatological dry season and little seasonality in rainfall".

However, in Figure 4 and Extended Figure 7 and 11, data from all six sites were presented for describing the relationships between soluble sugars (%) and minimum leaf water potential measured during the dry season (Figure 4). Similarly, Extended Figure 7 shows the relationships between leaf and branch water potential and soluble sugars (SS %) for all six sites. Again, Figure 4b and Extended Figure 11b only showed the distribution of seasonal shift of soluble sugars (ΔSS : %SSwet - %SSdry) for four sites. From these, it seems that at least soluble sugars were measured for both wet and dry season and for all the sites, but why only show the results for four sites for the dry season in majority of the results.

From Extended Data Figure 7, it seems that branch water potential for the sampled species were also measured, but in the Methods section there is not descriptions on how to measure branch water potential.

Extended Data Figure 3 and Figure 7: Are these relationships exist or hold among the species within each site? Or these relationships mainly caused by site differences of NSC concentration and functional traits (wood density, size, mortality)?

Extended Data Figure 7: Why one site (Man) is missing?

Extended Data Figure 10 and 11: Should be "Extended Data Figure 8 and 9, respectively"?

Extended Data Figure 3 and Figure 4: Why not present the relations between NSC and growth rate data? Is there possibility to calculate species-level stem growth rates from Forest inventory data, as sampled trees were from the RAINFOR network of forest permanent plots. In lines 46-47 in the abstract, the authors suggested that the NSC and its components may reflect differences in water status and seasonal patterns of productivity. Are seasonal NSCs related to seasonal growth rates?

Reviewer #3 (Remarks to the Author):

Understanding the role of drought on forest function in the Amazon and other tropical forests is critical to predicting the future role of these forests carbon sink in the global carbon cycle. Because of the massive carbon fluxes involved, uncertainty on how tropical forests will respond to potentially increased drought in the future limits our overall ability to project global climate change. This study assesses variation in NSC across a subcontinental scale for multiple tree species in an important area of the world, a region where NSC and its role in forest dynamics is extremely understudied. The assessment of the relative role of climate vs. taxonomy on non-structural carbohydrates using variance partitioning analysis is a novel approach for NSC dynamics that produced some really interesting results. The community-level approach for understanding NSC dynamics is also novel component of this study. Some specific comments follow.

Line 81: This statement implying that only one other study has assessed NSC in tropical forests and/or Amazonia is misleading. Technically, there may be only one study from a throughfall exclusion in the tropics (the cited Rowland et al.), but numerous studies of tropical tree NSC variation have been conducted in Central America, and also other tropical forests. Dunish and Puls (2003, *Journal of Applied Botany-Angewandte Botanik* 77:10–16) measured seasonal NSC variation in three Amazon tree species with greater frequency than the current study. (I acknowledge that this study is a poorly known and now defunct journal). A good source is the supplemental information for Martinez-Vilalta et al. 2016 (cited in this manuscript), or the data and metadata available here: <https://datadryad.org/stash/dataset/doi:10.5061/dryad.j6r5k>.

Line 91: All NSC analysis appears to have been conducted in one lab - even better. That would be

worth noting here.

Lines 109 and 405: (Psi)min is typically used to represent the minimum mid-day water potential observed over seasonal or annual measurements for a given species (and could also be calculated at the community level from the mean of multiple species), and is used to calculate hydraulic safety margin. But mid-day water potential was only measured once or twice for each species in this study. Change this symbol to (Psi)md, which is used for mid-day water potential.

Lines 128-198: These lines read more like a results section than a combined Results and Discussion, with a lot of detail, but minimal interpretation of results and placement of findings in a more general context. The organization of this section could be improved by restructuring paragraphs to include both results and discussion statements in each paragraph. If allowed, a simple solution could be to have separate Results and Discussion sections.

Line 159: Instead of "high degree of taxonomic control on leaf SS" I think you mean little variation of leaf SS across taxonomy. As stated, your point is unclear here. The phenomenon that SS is tightly regulated and varies much less than total NSC in plants and trees is well known, both in the tropics, and in other biomes. See Martinez-Villalta et al. 2016, a paper cited elsewhere in the manuscript, though not in this paragraph, for a good source that clarified this globally.

Lines 186, 208 and Fig 4. The difference between SS and %SS is confusing. When first examining Fig 4 it took me a while to figure out that this meant that %SS= [soluble sugars]/[NSCt]. It was confusing because [SS] is often reported as a percentage (although mg/g is used in this paper). Can this be stated as a proportion to avoid confusion? It's interesting that if you compare absolute changes in [SS] to the change in %SS from wet to dry season, you will not see consistent changes. Fig 3e shows that [soluble sugar] was slightly lower in the dry season at Ken, and higher in the wet season at Man. This is the opposite of results shown in Fig4b for these two sites. For branch data (Fig 3f) the change at Man in [SS] from wet to dry was a significant increase. It seems like the authors had to get a bit creative with the data in Fig 3 (pick one tissue and normalize to NSCt) to show the trend in Fig 4b that fits the conclusion. Leaf osmotic potential is not responding to the relative amount of [SS] compared to total NSC, it responds just to [SS]. Some discussion on this would improve the paper.

RESPONSE TO REVIEWERS COMMENTS

for the paper: “Non-structural carbohydrates mediate seasonal water stress across Amazon forests”.

We would like to thank the three reviewers for their helpful and constructive comments, which helped us to improve our manuscript. We are pleased with the positive assessments of our work, which recognise the breadth of the geographical sampling, the employment of standardised approaches over large spatial domains and the new insights that the study provides for understanding how NSC dynamics interacts with water availability across Amazon forests. Below we address all of the reviewers’ comments.

Reviewer #1 (Remarks to the Author):

1.1 This manuscript reports starch and simple sugar concentrations of terminal leaves and branches of 9-31 tree species from each of six Amazonian forests. Four of the sites are seasonal, with their dry seasons differing in severity and duration. In these 4 sites, leaves and terminal branches were sampled in both the wet and dry seasons. From two aseasonal sites without a dry season, sampling was done only during the wet season. This is a reasonable sampling design. The main scope was somewhat exploratory, but important because non-structural carbohydrates (NSC, the total of starch and simple sugar) are considered to be important for trees to cope with drought stress, which is widely recognized to be important for tree survival under the ongoing climate change. The manuscript summarizes a particularly nice dataset with broad geographical scope to allow generalization (six sites across Amazon), with standardized sampling.

Reply: We thank the reviewer for this positive and kind feedback on our work.

1.2 The methods involving tree climbing to sample terminal and well-exposed branches from the top of the canopy, and careful laboratory measurements, appear rigorous. The statistical approach is solid. The results demonstrate some interesting and novel discoveries, including overall lack of differences among sites despite their differences in seasonality, as well as strong taxonomic signals in leaf simple sugar concentrations. The results shown in Figure 4 are particularly interesting to plant ecologists and physiologists, as they suggest community-level differences in adaptive syndrome of the use of simple sugars for osmoregulation. A hypothesis that plant survival under drought depends on NSC storage was popular about 10-20 years ago. However, more recent studies and perspectives do not support that survival is a simple function depending on NSC concentration in plants. The results shown here are consistent with this current view, and highlight not the energy storage per se, but what is detected as simple sugar concentration may be important for drought survival. The simple sugars involved in osmoregulation, however, are unlikely glucose, fructose, or sucrose, as they interfere with metabolism. Instead, other types of oligosaccharides are involved in preventing leaves from wilting. So, the work would be stronger if they could report the identity of

the simple sugars that show differences among species, possibly with use of chromatography.

Reply: We thank the reviewer for the positive feedback about our work and for this insightful comment about identification of sugars specifically involved in osmoregulation. The objective of this study was to provide a general assessment of the role of NSCs and their constituent fractions (starch and soluble sugars) in mediating responses to water stress over large spatial scales and thus the identification of specific sugars was beyond the scope of the study. However, the identity of individual sugars would undoubtedly provide more information on species-specific responses to drought and future research should focus on identifying individual sugars involved in osmoregulation. In response to the reviewer's comment, we now include a section in the discussion on specific oligosaccharides that have been found to be important for osmoregulation in other studies (lines 239-243).

1.3 I find the very first sentence of the introduction to be a bit too ambiguous or too general. The question here is the energy storage, although it is impossible to separate what are NSC in active ongoing metabolism vs. storage. Starch and simple sugars have different functions with this regard.

Reply: Based on the reviewer's point, we have now rewritten this sentence to emphasise the fact that NSCs are important stores of energy. (Lines 58-60).

1.4 Any species were shared among sites? In the variance partitioning, species are important, but if there was any species shared between two or more sites, how they differ among sites can be informative.

Reply: We agree that examination of NSC dynamics in species found across multiple sites can be very informative and we have now made changes in the article to accommodate this, both in the discussion (lines 150-151; 223-225) and through new figures in the Supplementary Information (SI Figs. 3-6, 11 & 12). In total, nine species occur across more than one site in our 'dry' season dataset (four sites; Supplementary Fig. 11 & 12) and 22 species occur across more than one site in the 'wet' season dataset (six sites; Supplementary Fig. 3-6). Species common across sites generally show similar concentrations of NSC in the wet season. However, in the dry season we observed higher concentrations of soluble sugar in the drier site. These results are in line with the variance partitioning results, which shows that site has a greater importance in the concentration of soluble sugar during the dry season, but not in the wet season (Fig. 2).

1.5 *Leaf phenology can be important, but the manuscript does not mention anything about the phenology of the sampled species. Are they all evergreen species, and if so, when are the leaves produced and become fully expanded? Even in forests with limited seasonality, tree species show distinctive leaf production / replacement phenology.*

Reply: We agree that NSC seasonal fluctuation and differences in allocation to starch or soluble sugars could be coordinated with leaf phenology. Unfortunately, even basic leaf phenology data for Amazon species (e.g. whether a species is deciduous or not) is scarce, with information on the timing of leaf loss and flush being even rarer. In our study, we were able to source leaf phenology information for 40 out of 82 species, which are as follows, 25 evergreen, eight semi-deciduous and seven deciduous species (species phenology is now added to the Supplementary Notes). We tested for differences in NSC concentrations among the different phenological groups and found that evergreen species have higher total leaf NSC_T ($p = 0.023$) and SS ($p = 0.012$) in the wet season than semideciduous/deciduous species and now updated the supplementary information to reflect this analysis (Supplementary Fig. 15 & 16). We have now added a sentence to the main text (lines 155-159) which describes this result, noting the caveat that we did not have phenological information for a large proportion of our species. Previous site-specific studies (e.g. the study by Würth *et al.* 2005 in Panama) have also not found strong evidence of differences between species from different phenological groups in 16 Panamanian species.

(Würth, M. K. R., Peláez-Riedl, S., Wright, S. J. & Körner, C. (2005). Non-structural carbohydrate pools in a tropical forest. *Oecologia* **143**, 11–24)

1.6 *Line 388 – simple sugars are expressed as glucose equivalent, but it may contain various sugars other than glucose, fructose, and sucrose. Especially, plant leaves use sugars other than these three for osmotic regulations.*

To sum up, I like the manuscript very much, and overall the authors have done a great job in writing clearly. My main concern is that we don't know the identity of the simple sugars (SS) that show strong species and family-level signals (Figure 2), site differences (Figure 4a), and seasonal responses at the species-level (Figure 4b). I agree with the authors' interpretation that SS differences reflect species-level differences in the use of osmoregulating sugars (in leaf cell vacuoles to avoid leaf wilting under water stress). But, the chemicals that cause species and seasonal differences in leaf SS concentrations are unlikely to be glucose, fructose, or sucrose. The authors may think that knowing their identity is beyond the scope of their work, but at least they have to discuss what are potential chemicals that act as osmoregulators.

Reply: This point is equivalent to the first point made by the reviewer (Comment 1.2). As highlighted in our response to Comment 1.2, we now add a section to the discussion (lines 239-243) in which we discuss the importance of identifying osmoregulatory sugars and indicate what some of these sugars might be, based on the wider literature on the topic.

Reviewer #2 (Remarks to the Author):

Comments on “Non-structural carbohydrates mediate seasonal water stress across Amazon forests”

General comments:

2.1 Non-structural carbohydrates (NSC) play an important role in tolerance to water stress and may mediate drought-induced mortality. This paper explored variabilities of total NSC and its components (soluble sugars and starch) across 83 canopy tree species in six Amazonian sites across a broad precipitation gradient. Total NSC concentrations in both leaves and branches were found similar across sites in wet season, but varied much more across sites during the dry season. At the community scale, leaf soluble sugars were found strongly linked to minimum leaf water potential. In drier sites, leaf soluble sugars increased during the dry season for almost all species, suggesting that leaf soluble sugars play a role in maintaining hydraulic status during water deficits in drier sites.

From the precipitation distribution map (Figure 1), as well as from Supplementary Figure 5 and 6, site Ken is the driest site with longest dry season, while Fec site range as the second driest site. However, the seasonal variation of total NSC, starch and soluble sugars (SS) showed opposite patterns, especially for leaves (Figure 3a, e, f), as dry season leaf NSC and SS were lower for Ken site, but high for Fec site as compared to those in wet season. At the less dry site Fec, absolute soluble sugars were higher than other sites for both wet and dry seasons, what are the possible reasons?

Reply: The reviewer asks us to elaborate on the differences in patterns observed at the two driest sites (Ken and Fec) in our network. Despite trees in both sites decreasing their leaf starch concentrations during the dry season, there is an increase in leaf soluble sugars in Fec, but not in Ken, where soluble sugar concentrations remained similar to those found in the wet season (Fig. 3). We believe that the differences between these sites can be explained through the substantial dry season source limitation in Ken, which is by far the driest site in our gradient. Monthly growth measurements from this site show that during the dry season stem growth in this site is reduced to near zero levels (Girardin *et al.* 2016). In Fec, however, where the dry season is not of the same intensity as in Ken, we expect source limitation to be weaker and indeed the increase in NSC_T observed at this site in the dry season suggests no real source limitation. In Fec, we expect metabolic rates and

growth to be higher in the dry season than in Ken and that SS might increase in the dry season as a result of this, as additional osmoregulatory requirements are coupled with greater demand to maintain metabolic rates. These ideas were in the original version but have been made clearer in this new version of the manuscript (lines 185-194).

(Girardin, C. A. J., *et al.*, (2016). Seasonal trends of Amazonian rainforest phenology, net primary productivity, and carbon allocation, *Global Biogeochem. Cycles*, 30, 700–715)

2.2 In the Results and Discussion (Line 215-220): The authors described these mixed results like: “In FEC, all species without exception increased the absolute leaf SS concentration in the dry season and all species except one depleted leaf starch stores. In KEN, dry season source limitation meant that several species did not increase absolute leaf SS concentrations and instead relied on larger depletions of leaf starch stores to maintain dry season SS at similar levels to the wet season (Figure 3).” Is this a site specific pattern? The opposite seasonal shift between Ken and Fec site reduce the general picture and make the results more difficulty to be explained.

Reply: We agree with the reviewer that these sentences were hard to follow. Reviewer three has also pointed out that some parts of the results/discussion were hard to follow. Because of your comments we have now restructured the results and discussion session. The discussion section mentioned by the reviewer has been re-written (lines 185-194) and now reads: “In the two driest sites in our network, Ken and Fec, we find strong evidence of mobilisation of starch reserves to SS. In Ken, the driest site evaluated, leaf starch reserves declined by 81% ($p < 0.001$) in the dry season while leaf SS concentrations remained unchanged ($p = 1$), despite a 43% reduction in leaf NSC_T (Fig 3, $p = 0.019$). In Fec, the second driest site evaluated, leaf starch concentrations also decreased markedly in the dry season (72% reduction; $p < 0.001$). However, in this site, significant increases in leaf (32% increase; $p < 0.001$) and branch (48% increase; $p < 0.001$; Fig. 2) SS were observed, with an overall increase in leaf NSC_T in the dry season. The reduction of leaf NSC_T in Ken but not in Fec may be attributed to stronger source limitation in the driest Ken site (Extended Data Fig. 5)”.

2.3 Moreover, Figure 4 showed that Fec site always have the highest (see also Extended Data Figure 11) leaf and branch SS (% of leaf NSC corresponding to soluble sugars), but the distribution of of delta SS (SS_{wet} – SS_{dry}) at Fec site stay in between the driest site (Ken) and other wetter sites (Man and Tam). Is it possible use the delta SS (soluble sugar differences between wet and dry season) as a parameter for describing the relationships between leaf/branch water potential and NSC (total NSC, starch, and SS).

Are there any relationships between leaf water potential and total NSC and starch concentrations?

Reply: The reviewer raises an interesting question about the relationship between leaf water potential and specific NSC fractions. In line with a recent study (Guo *et al.*, 2020), we found that absolute leaf starch concentrations were strongly positively to water potential but that absolute leaf SS concentrations were not. Thus, across sites the relationship between the proportion of leaf NSC_T in the form of soluble sugars (now referred as SS:NSC_T) and water potential shown in Figure 4a is more strongly driven by starch depletion (Fig. 4c) than by absolute increases in SS. We have now added these results into the main text lines 214-218 and in the Supplementary Information Fig. 15. (Guo, J. S., Gear, L., Hultine, K. R., Koch, G. W. & Ogle, K. (2020). Non-structural carbohydrate dynamics associated with antecedent stem water potential and air temperature in a dominant desert shrub. *Plant Cell Environ.* doi:10.1111/pce.13749)

2.4 Another concern is about the unbalance presentations of the data from full six study sites (named Ken, Fec, Man, Tam, Alp, Suc). From Figure 2, Figure 3 and Extended Data Figures 2, it seems that the experimental design did not consider sampling and NSC measurements for the wetter sites (Alp and Suc) during the dry season (Figure 2c, g; Figure 3, Extended Data Figure 2 c, d, g, h). In the Methods section, the authors did not indicated that NSC sample were not collected and measured for Alp and Suc sites during the dry season. Moreover, in the Methods section Lines 411-413, the authors indicated that “Water potential data collection took place in what is typically the driest time of the year in each sampling plot expect in the ALP and SUC sites where there is no climatological dry season and little seasonality in rainfall”. However, in Figure 4 and Extended Figure 7 and 11, data from all six sites were presented for describing the relationships between soluble sugars (%) and minimum leaf water potential measured during the dry season (Figure 4). Similarly, Extended Figure 7 shows the relationships between leaf and branch water potential and soluble sugars (SS %) for all six sites. Again, Figure 4b and Extended Figure 11b only showed the distribution of seasonal shift of soluble sugars (delta SS: %SSwet - %SSdry) for four sites. From these, it seems that at least soluble sugars were measured for both wet and dry season and for all the sites, but why only show the results for four sites for the dry season in majority of the results.

Reply: It is conventional for Amazonian studies to consider the dry season as months where precipitation is lower than 100 mm (Sombroek, 2001). In the Alp and Suc sites, there is no period with precipitation lower than 100 mm month⁻¹ (Fig. 1, Extended Data Fig. 1). Thus, due to the absence of a marked dry season and the logistic difficulty to conduct sampling in these sites, NSC sampling and leaf water potential were taken only once in the Alp and Suc sites (lines 100-104; 110-112; 249-251; 484-486 and Extended

data Tab. 1 and Extended Data Fig. 1). As reviewer one highlights, this is a reasonable sampling design (Point 1.1). As they do not experience a dry season, as conventionally defined, we did not include these two sites in the comparison of dry season differences in the various NSC metrics across sites. However, we are confident that the minimum water potential values measured in these sites are representative of the minimum values that they would experience throughout the course of the year and thus include them in the analyses relating water potential to NSC metrics.

(Sombroek, W. (2001) Spatial and Temporal Patterns of Amazon Rainfall. *AMBIO A J. Hum. Environ.* **30**, 388–396)

2.5 From Extended Data Figure 7, it seems that branch water potential for the sampled species were also measured, but in the Methods section there is not descriptions on how to measure branch water potential.

Reply: Water potential was measured only in leaves and not branches. We have now clarified the methods section to make this point clearer (lines 478-488) and also updated figure legends to clarify this point.

2.6 Extended Data Figure 3 and Figure 7: Are these relationships exist or hold among the species within each site? Or these relationships mainly caused by site differences of NSC concentration and functional traits (wood density, size, mortality)?

Reply: To construct these figures we used the NSC data from the wet season, when there is no evident water stress. As there is little difference in NSC concentrations between the same species sampled in different sites in the wet season (Fig. 1 and Extended Data Fig. 2), for this figure we use the average value per species, and not species per site. This should be very clear from the figure legends. Thus, the relationship should be driven by species characteristics and not by sampled sites.

2.7 Extended Data Figure 7: Why one site (Man) is missing?

Reply: Due to logistical limitations predawn leaf water potential was not measured in the Man site (lines 630-631). We have now added this information in the methods section, lines 487-488.

2.8 Extended Data Figure 10 and 11: Should be “Extended Data Figure 8 and 9, respectively”?

Reply: Thank you for reading our manuscript carefully and calling our attention to this. We have now corrected the figure numbering.

2.9 *Extended Data Figure 3 and Figure 4: Why not present the relations between NSC and growth rate data? Is there possibility to calculate species-level stem growth rates from Forest inventory data, as sampled trees were from the RAINFOR network of forest permanent plots. In lines 46-47 in the abstract, the authors suggested that the NSC and its components may reflect differences in water status and seasonal patterns of productivity. Are seasonal NSCs related to seasonal growth rates?*

Reply: We thank the reviewer for raising this interesting point. When we refer to the seasonal patterns of productivity we are talking about those inferred from satellite-derived EVI (Enhanced Vegetation Index) (Extended Data Fig. 5). RAINFOR plot re-censuses typically span 2-3 years and do not allow for seasonal resolution of growth patterns. However, we do concur with the reviewer that it is indeed important to add the relationship between NSC and species-level growth rates and have included this in Extended Data Fig. 3-4. For this analysis, we used the mean growth rate (cm/year) data at the species level calculated by (Coelho de Souza *et al.*, 2016).

(Coelho de Souza, F. *et al.* (2016). Evolutionary heritage influences Amazon tree ecology. *Proc. R. Soc. B Biol. Sci.* 283).

Reviewer #3 (Remarks to the Author):

3.1 *Understanding the role of drought on forest function in the Amazon and other tropical forests is critical to predicting the future role of these forests carbon sink in the global carbon cycle. Because of the massive carbon fluxes involved, uncertainty on how tropical forests will respond to potentially increased drought in the future limits our overall ability to project global climate change. This study assesses variation in NSC across a subcontinental scale for multiple tree species in an important area of the world, a region where NSC and its role in forest dynamics is extremely understudied. The assessment of the relative role of climate vs. taxonomy on non-structural carbohydrates using variance partitioning analysis is a novel approach for NSC dynamics that produced some really interesting results. The community-level approach for understanding NSC dynamics is also novel component of this study. Some specific comments follow.*

Reply: We really appreciate these comments from the reviewer on the novelty of the work. Thank you.

3.2 *Line 81: This statement implying that only one other study has assessed NSC in tropical forests and/or Amazonia is misleading. Technically, there may be only one study from a throughfall exclusion in the tropics (the cited Rowland *et al.*), but numerous studies of tropical tree NSC variation have been conducted in Central America, and also other tropical forests. Dunish and Puls (2003, - -Angewandte*

Botanik 77:10–16) measured seasonal NSC variation in three Amazon tree species with greater frequency than the current study. (I acknowledge that this study is a poorly known and now defunct journal). A good source is the supplemental information for Martinez-Vilalta et al. 2016 (cited in this manuscript), or the data and metadata available here: <https://datadryad.org/stash/dataset/doi:10.5061/dryad.j6r5k>.

Reply: We thank the reviewer for this constructive comment and for sharing this very interesting but little known study that we were unaware of. We have now adjusted the language in this sentence and cited the study referred to (lines 80-85).

3.3 Line 91: All NSC analysis appears to have been conducted in one lab - even better. That would be worth noting here.

Reply: The laboratory standardisation is indeed an important element of our study design which we had not explicitly included in the methods. We have now highlighted this in lines 91-93.

3.4 Lines 109 and 405: (Psi)min is typically used to represent the minimum mid-day water potential observed over seasonal or annual measurements for a given species (and could also be calculated at the community level from the mean of multiple species), and is used to calculate hydraulic safety margin. But mid-day water potential was only measured once or twice for each species in this study. Change this symbol to (Psi)md, which is used for mid-day water potential.

Reply: We agree with the reviewer that it would be better changing (Psi) minimum to (Psi) midday and have made this change throughout the text and figures.

3.5 Lines 128-198: These lines read more like a results section than a combined Results and Discussion, with a lot of detail, but minimal interpretation of results and placement of findings in a more general context. The organization of this section could be improved by restructuring paragraphs to include both results and discussion statements in each paragraph. If allowed, a simple solution could be to have separate Results and Discussion sections.

Reply: Following the suggestion of the reviewer, we have now substantially restructured this section to be less descriptive and to integrate discussion and results elements more clearly.

3.6 Line 159: Instead of “high degree of taxonomic control on leaf SS” I think you mean little variation of leaf SS across taxonomy. As stated, your point is unclear here. The phenomenon that SS is tightly regulated and varies much less than total NSC in plants and trees is well known, both in the tropics, and in other biomes. See Martinez-Vilalta et al. 2016, a paper cited elsewhere in the manuscript, though not in this paragraph, for a good source that clarified this globally.

Reply: We agree with the reviewer that this sentence was unclear, and the reviewer's confusion here reflects this lack of clarity. The point we wanted to make is actually different to the Reviewer's interpretation and really relates to the fact that taxonomy explains much more of the variance in SS than it does in starch, in both wet and dry seasons. We have now changed this sentence to clarify this point (lines 159-165).

3.7 Lines 186, 208 and Fig 4. The difference between SS and %SS is confusing. When first examining Fig 4 it took me a while to figure out that this meant that %SS= [soluble sugars]/[NSC_T]. It was confusing because [SS] is often reported as a percentage (although mg/g is used in this paper). Can this be stated as a proportion to avoid confusion?

Reply: We agree that “Soluble Sugars (%)”, as shown in the axis of the figure 4a could bring some confusion as in some cases NSC can be represented as % of dry weight, e.g. SS (% d.w). To avoid ambiguity, we change figures axes to “Soluble Sugar Fraction (SS:NSC_T)”, and use this terminology throughout the manuscript.

3.8 It's interesting that if you compare absolute changes in [SS] to the change in %SS from wet to dry season, you will not see consistent changes. Fig 3e shows that [soluble sugar] was slightly lower in the dry season at Ken, and higher in the wet season at Man. This is the opposite of results shown in Fig4b for these two sites. For branch data (Fig 3f) the change at Man in [SS] from wet to dry was a significant increase. It seems like the authors had to get a bit creative with the data in Fig 3 (pick one tissue and normalize to NSC_T) to show the trend in Fig 4b that fits the conclusion. Leaf osmotic potential is not responding to the relative amount of [SS] compared to total NSC, it responds just to [SS]. Some discussion on this would improve the paper.

Reply: We believe the reviewer's point here is similar to the point made by Reviewer 2 in comment 2.3 where we are asked to explore the relationships between water potential and absolute NSC fractions (starch, soluble sugars). We have now conducted new analyses of these relationships and update both the main text and supplementary information to reflect this (see response to comment 2.3). The SS:NSC_T ratio reflects the relative prioritisation of SS vs. starch as an NSC store. We show clearly in Fig. 4c that during the dry season, absolute starch stores decline markedly along our water potential gradient, whereas absolute SS concentrations do not (i.e. they are prioritised). We have now developed our discussion to make these points clearer.

REVIEWERS' COMMENTS

Reviewer #1 (Remarks to the Author):

Overall, I am satisfied with how the authors responded to my comments in the previous round or review, as summarized in the response letter.

Minor points:

Line 152-155: It is a very long sentence. Break down to two sentences? Also, there is no need to start the sentence with "we find...".

Line 184 : , reflecting the dual control of plant water status and the seasonality of plant productivity. – I don't understand. Rephrase?

Line 191-196. "The reduction of leaf NSC in Ken but not in Fec may be attributed to stronger source limitation in the driest Ken site." This sentence is too speculative. The data support is the stronger EVI seasonality, which could simply reflect a greater proportion of deciduous tree crowns. Productivity at the ecosystem (canopy) level may be more depressed, but in this study, which is focusing on tree-level (or species-level) response, this statement is still a bit too speculative.

Line 196-197: "depletion of branch starch may be due to enhanced growth facilitated by branch SS concentrations." I do not agree with the logic here. Seasonal shoot extension may be associated with higher SS concentration, but it is misleading to state that it is "facilitated".

Line 218: "important prioritization of SS in leaves in the period of water stress". The term "important prioritization" should be replaced with more objective term, such as "notable increase". It is not clear priority of what over what. It is odd to state SS has priority over starch. Rather, stored starch is simply converted to SS.

Line 228. "Our observation of pervasive prioritization of foliar SS relative to starch". For the same reason as my previous comments, more precise statement, such as "The increased foliar SS relative to starch in the drier sites"...There is also no need to start the sentence with "Our observation".

Line 233. Again, I do not like the term "an active prioritization of SS". It can be simply "an increase of SS".

Line 263. "forests which are less adapted to water deficit" should be "tree species that are less adapted to strong seasonal drought". Forests are not the unit of evolutionary adaptation. Your study's focus is tree species adaptation (which is well supported by the strong taxonomic signal as you show).

Line 106 and line 461: According to the method description, the enzymatic (invertase) reaction was used to break down fructose and sucrose to get the total SS concentration. I am not sure whether oligosaccharides may be broken down by the invertase derived from *Saccharomyces*. Glucose and sucrose are poor molecules as osmoregulators, because they interfere with leaf metabolism (and so their concentrations are likely to be regulated). While I understand why you may equate simple sugar to the total of glucose, sucrose and fructose, I still suggest, simple sugars (i.e., oligosaccharides such as glucose, sucrose, fructose etc.) , unless you have evidence supporting that the invertase reaction does not break down other oligosaccharides.

This may be a matter of individual stylistic preference, but I find the authors use too many "we find", "our observation", etc. I personally prefer neutral phrases to state the facts without emphasizing "we".

Reviewer #2 (Remarks to the Author):

I think the authors have made sufficient revision on the manuscript and improved the quality of writing. I suggest acceptance as the current form.

RESPONSE TO REVIEWERS COMMENTS

for the paper: “Non-structural carbohydrates mediate seasonal water stress across Amazon forests”

Reviewer #1 (Remarks to the Author):

1.1 Overall, I am satisfied with how the authors responded to my comments in the previous round or review, as summarized in the response letter.

Reply: We were pleased to know that we have been able to respond to the reviewer’s comments satisfactorily.

Minor points:

1.2 Line 152-155: It is a very long sentence. Break down to two sentences? Also, there is no need to start the sentence with “we find....”.

Reply: As suggested by the reviewer, we have removed “we find” and have shortened the sentence.

Line:

1.3 Line 184:, reflecting the dual control of plant water status and the seasonality of plant productivity. – I don’t understand. Rephrase?

Reply: We agree that the sentence could be improved, and we have removed the second clause to avoid ambiguity. The potential controls of water stress and growth on NSC dynamics are discussed later in the paragraph.

1.4 Line 191-196. “The reduction of leaf NSC in Ken but not in Fec may be attributed to stronger source limitation in the driest Ken site.” This sentence is too speculative. The data support is the stronger EVI seasonality, which could simply reflect a greater proportion of deciduous tree crowns. Productivity at the ecosystem (canopy) level may be more depressed, but in this study, which is focusing on tree-level (or species-level) response, this statement is still a bit too speculative.

Reply: We disagree with the reviewer that this statement is too speculative. We have been careful to formulate the phrase in a way that provides a potential mechanism for the differences between the two sites but which does not make conclusive assertions. The reviewer suggests that the seasonal EVI data for Kenia might reflect a greater degree of deciduousness than a limitation of photosynthesis. However, the number of deciduous species found on our KEN site (2/9) is of similar proportion to those in the FEC site (3/14) and we think it is unlikely that differences in phenology drive this signature.

1.5 Line 196-197: “depletion of branch starch may be due to enhanced growth facilitated by branch SS concentrations.” I do not agree with the logic here. Seasonal shoot extension may be associated with higher SS concentration, but it is misleading to state that it is “facilitated”.

Reply: We thank the reviewer for raising this point and agree that the sentence may be misleading. We have restructured the sentence and no longer use the word ‘facilitated’.

1.6 Line 218: *“important prioritization of SS in leaves in the period of water stress”*. The term *“important prioritization”* should be replaced with more objective term, such as *“notable increase”*. It is not clear priority of what over what. It is odd to state SS has priority over starch. Rather, stored starch is simply converted to SS.

Reply: In response to the reviewer’s suggestion, we no longer use the term ‘prioritization’.

Line

1.7 Line 228. *“Our observation of pervasive prioritization of foliar SS relative to starch”*. For the same reason as my previous comments, more precise statement, such as *“The increased foliar SS relative to starch in the drier sites”* ... There is also no need to start the sentence with *“Our observation”*.

Reply: We changed this sentence as suggested by the reviewer.

Line

1.8 Line 233. Again, I do not like the term *“an active prioritization of SS”*. It can be simply *“an increase of SS”*.

Reply: We changed this sentence to accommodate the reviewer suggestion.

Line

1.9 Line 263. *“forests which are less adapted to water deficit”* should be *“tree species that are less adapted to strong seasonal drought”*. Forests are not the unit of evolutionary adaptation. Your study’s focus is tree species adaptation (which is well supported by the strong taxonomic signal as you show).

Reply: We agree with the reviewer comment that forests are not the unit of evolutionary adaptation. We modified this sentence to accommodate the reviewer’s suggestion.

Line

1.10 Line 106 and line 461: According to the method description, the enzymatic (invertase) reaction was used to break down fructose and sucrose to get the total SS concentration. I am not sure whether oligosaccharides may be broken down by the invertase derived from *Saccharomyces*. Glucose and sucrose are poor molecules as osmoregulators, because they interfere with leaf metabolism (and so their concentrations are likely to be regulated). While I understand why you may equate simple sugar to the total of glucose, sucrose and fructose, I still suggest, simple sugars (i.e., oligosaccharides such as glucose, sucrose, fructose etc.)”, unless you have evidence supporting that the invertase reaction does not break down other oligosaccharides.

Reply: We have modified both sentences as suggested by the reviewer.

1.11 This may be a matter of individual stylistic preference, but I find the authors use too many *“we find”*, *“our observation”*, etc. I personally prefer neutral proses to state the facts without emphasizing *“we”*.

Reply: We understand that some people may prefer more neutral proses, however as it refers to the personal style of writing, we decided to keep the text as it is.

Reviewer #2 (Remarks to the Author):

2.1 I think the authors have made sufficient revision on the manuscript and improved the quality of writing. I suggest acceptance as the current form.

Reply: We are happy that the changes we have made in the manuscript made the reviewer satisfied and thanks he has suggested the manuscript acceptance.